# Comparison of Free, Esterified, and Insoluble-Bound Phenolics and Their Bioactivities in Three Organs of *Lonicera japonica* and *L. macranthoides*

**DOI:** 10.3390/molecules24050970

**Published:** 2019-03-09

**Authors:** Miao Yu, Lingguang Yang, Qiang Xue, Peipei Yin, Liwei Sun, Yujun Liu

**Affiliations:** College of Biological Sciences and Biotechnology, Beijing Forestry University, Qinghuadonglu No. 35, Haidian District, Beijing 100083, China; 13261388622@163.com (M.Y.); yanglingguangxdjqz@163.com (L.Y.); xuetian20130607@163.com (Q.X.); happy62889@126.com (P.Y.)

**Keywords:** *Lonicera japonica*, *Lonicera macranthoides*, phenolics, antioxidant activity, DNA protection, UPLC-DAD–QTOF-MS/MS

## Abstract

Dried flower buds of *Lonicera japonica* and *L. macranthoides* have long been used as herbs in numerous Chinese traditional medicines. Comparisons of three phenolic fractions (i.e., free, esterified, and insoluble-bound phenolics) in three different organs (i.e., flower, leaf, and stem) of the two species revealed that the free phenolics were the highest in terms of total phenol and total flavonoid content, composed of the most numerous phenolics and flavonoids; thus, they exhibited the most excellent antioxidant activities (2,2-diphenyl-1-picrylhydrazyl (DPPH), 2,2′-azino-bis(3-ethylbenzthiazoline-6-sulfonate) (ABTS), and oxygen radical absorbance capacity (ORAC)), as well as protective effects on DNA damage induced by free radicals. In identical free and esterified phenolics of a same organ, higher contents and bioactivities were observed in *L. macranthoides* than in *L. japonica*. Phenolics identified by ultra-performance liquid chromatography with a diode array detector, alongside tandem mass spectrometry coupled with a quadrupole time-of-flight mass spectrometer (UPLC-DAD–QTOF-MS/MS) mainly included chlorogenic acid and its five derivatives, three flavonoids that were only found in the free phenolic fraction and closely correlated with its bioactivity, and caffeic acid that was the major contributor to antioxidant activity of the esterified and insoluble-bound phenolic fractions. It was, thus, concluded that, like *L. japonica*, *L. macranthoides*, which was underestimated since being separately listed by the 2010 edition of the Chinese Pharmacopoeia, is also a good (and better) herbal medicine.

## 1. Introduction

As one of the most important genera in the family Caprifoliaceae, many *Lonicera* species are widely recognized in traditional Chinese medicine, as well as for their use in health-promoting beverages and for the treatment of sores, acute fever, headache, carbuncles, acute rheumatoid arthritis, swelling, upper respiratory tract infections, diabetes, and throat inflammations [1]. In the Chinese Pharmacopoeia published in 2010, dried flower buds of *Lonicera macranthoides*, *L. hypoglauca*, *L. confusa*, and *L. fulvotomentosa* were recorded and are collectively known as “Shanyinhua”, while dried buds of *L. japonica* were listed individually as “Jinyinhua”, despite in fact possessing nearly the same flavor, meridian tropism, functions, and indications [1]. Since then, only Jinyinhua was deemed to be one of the specific original materials by most Chinese prescriptions and beverage formulae (e.g., the popular Chinese herbal tea “Jia Duo Bao”), which is also why the market price of Jinyinhua is much higher than that of the four Shanyinhua, leading to the latter being discriminated against [2,3].

Actually, the commercial value of Jinyinhua in herbal medicine and food trading markets increased over 400% in recent years, while that of the four Shanyinhua decreased [4]. As congeneric species of *L. japonica* (i.e., the Jinyinhua species), the four Shanyinhua species as mentioned above, especially *L. macranthoides*, were widely cultivated in southern China and were also used in processing a number of traditional Chinese medicines and beverages before the differential treatment by the 2010 edition Chinese Pharmacopoeia. Such issues reflect an urgent need to investigate the biological activities of the Shanyinhua species for the purpose of their further promotion and application.

Previous studies found that major functional compositions of *L. japonica* include phenolics, essential oils, triterpenoid saponins, iridoids, and organic acids, among more than 150 isolated chemical compounds [5]. Therein, phenolics such as luteolin, caffeic acid, caffeoylquinic acid isomers, and dicaffeoylquinic acid isomers were believed to be the major active ingredients of *L. japonica*, possessing outstanding pharmacological effects such as antibacterial, antioxidant, anti-inflammatory, anti-diabetic, anti-encystment, anti-amoebicidal, and anticancer activities [5,6,7,8,9,10,11]. Pharmacological studies revealed that phenolics can donate hydrogens or electrons to scavenge a number of radicals (such as singlet oxygen, superoxide anion, hydroxyl radical, the 2,2-diphenyl-1-picrylhydrazyl (DPPH) radical, and the 2,2′-azino-bis(3-ethylbenzothiazoline-6-sulfonic acid) (ABTS) radical cation) particularly associated with a couple of diseases including cancer, diabetes, and cardiovascular disease [12,13,14]. In contrast to intensive researches on *L. japonica*, studies on *L. macranthoides* are limited. A few reports on *L. macranthoides* focused on identifications of chemical substances, pointing out its potential pharmaceutical properties, but lacking quantitative analyses, as well as comparative investigations with *L. japonica* [1,15].

Phenolics, according to their existing forms in plants, are generally divided into three classes, i.e., free, esterified, and insoluble-bound phenolics. Free phenolics, as well as esterified phenolics that are conjugated to sugars and low-molecular-mass components, are extractable by a solvolytic solution such as water, methanol, ethanol, and acetone, while insoluble-bound phenolics are covalently bound to the cell-wall structural component and are commonly extracted by alkalization of the residue after the extraction of free and esterified phenolics [16,17,18]. Chen et al. [19] reported that the contents of esterified and insoluble-bound phenolics were significantly higher (*p* < 0.05) than that of free phenolics in leaves of *Artocarpus heterophyllus* and *Averrhoa carambola*. Similarly, blackberry, black raspberry, and blueberry also contained relatively higher proportions of esterified and insoluble-bound phenolics [20]. The contents of insoluble-bound phenolics in oat (*Avena sativa*) and blue maize nejayote even reached up to 88.04% and 91.05% of the total phenolics, respectively [21,22]. To the best of our knowledge, no reports were carried out on the free, esterified, and insoluble-bound phenolics of both *L. japonica* and *L. macranthoides*. Therefore, the present work investigated free, esterified, and insoluble-bound phenolics extracted from three different organs of both species by ultra-performance liquid chromatography with a diode array detector, alongside tandem mass spectrometry coupled with a quadrupole time-of-flight mass spectrometer (UPLC-DAD–QTOF-MS/MS), and, for the first time, compared their antioxidant activities.

## 2. Results and Discussion

### 2.1. Differences in Contents of Three Phenolic Fractions in L. japonica and L. macranthoides

Free, esterified, and insoluble-bound fractions of total phenols in *L. japonica* and *L. macranthoides* are shown in Table 1 (top). Free phenolics were concentrated in the flower of *L. japonica* (150.44 μmol gallic acid equivalent per gram of dry weight (μmol GAE/g DW)) and the leaf of *L. macranthoides* (365.83), while esterified phenolics were highest in the flowers of both species (38.80 and 77.04, respectively). Insoluble-bound phenolics were concentrated in the leaf of *L. japonica* (29.97) and the stem of *L. macranthoides* (16.10). In all three organs, free phenolics were all significantly higher than either the esterified or the insoluble-bound phenolics (see averages in Table 1), suggesting that free phenolics were the major form of total phenols in both *Lonicera* species. Furthermore, with the exception of insoluble-bound phenolics being a little higher on average in *L. japonica* than in *L. macranthoides* (14.07 vs. 13.57), free and esterified phenolics on average in all three organs (Table 1, top; data colored in blue) and total phenols in each of the three organs (Table 1, top) were much lower in *L. japonica* than in *L. macranthoides*, indicating that *L. macranthoides* is certainly superior to *L. japonica* with regard to the contents of different fractions, as well as total phenols.

As shown in Table 1 (bottom), free flavonoids were the highest also in the flower of *L. japonica* (60.64 μmol RE/g DW) and the leaf of *L. macranthoides* (166.59), whereas esterified and insoluble-bound flavonoids maximized in flowers (16.39 vs. 33.96) and leaves (12.84 vs. 6.45) of both species, respectively. In all three organs, as also indicated by the averages, free flavonoids were the highest, followed by esterified and insoluble-bound flavonoids, implying that free flavonoids were also the major form of total flavonoids in both species. Furthermore, total flavonoids exhibited exactly the same pattern as that for the total phenols with regard to the three fractionated flavonoids in all three organs (Table 1, bottom; even the insoluble-bound flavonoids as an exception was the same) and total flavonoids in each of the three organs (Table 1, bottom), further indicating that *L. macranthoides* is also superior to *L. japonica* with respect to contents of different flavonoid fractions, as well as total flavonoids. These results are surprising as it was discriminated against in the medicinal market due to the reclassification of the Chinese Pharmacopoeia 2010 edition, as pointed out in Section 1.

All the above results demonstrated that contents of free phenolics were the highest among the three fractions and that *L. macranthoides* was superior to *L. Japonica* in the free and esterified fractions (Table 1). To be more specific, among the three different organs of the two species, total phenols and total flavonoids of each organ ranged from 23.71 to 439.13 μmol GAE/g DW with a biggest difference of 18.52-fold and the exact same order for total phenols and total flavonoids as follows: leaf of *L. macranthoides* > flower of *L. macranthoides* > flower of *L. Japonica* > leaf of *L. Japonica* > stem of *L. macranthoides* > stem of *L. Japonica* (Table 1).

Higher proportions of the free phenolic fraction in total phenols wwew also found in the leaves of pomelo, longan, and litchi [19], and in hyacinth bean, mung bean, pearl bean, *Phaseolus calcaratus*, and Semen Dolichoris Album [23], whereas, in black bean, chickpea, cow gram, flower waist bean, kidney bean, red bean, red kidney bean, and spring bay bean, contributions of insoluble-bound phenolics were higher than those of free and esterified phenolic fractions [23]. Free, conjugated, and bound phenolic acids of seven commonly consumed vegetables, including kidney bean, cow pea, snow pea, hyacinth bean, green soy bean, soybean sprouts, and daylily were characterized [24]. Ambigaipalan et al. [25] demonstrated that the free phenolic fraction of the leathery outer skin from pomegranate peel possessed higher total phenols than esterified and insoluble-bound phenolic fractions, while the esterified fractions of mesocarp (spongy part of peel) and divider membrane were relatively higher. As for flavonoids, Min et al. [26] found that the major contributors to total flavonoids were free flavonoids in red and purple bran types of rice, but the whole-grain rice contained a significant amount of bound flavonoids. Alshikh et al. [27] studied five different varieties of lentils and found that most soluble total flavonoids (free plus esterified) were higher than the fraction containing insoluble-bound flavonoids. All these results also provide evidence that there are various forms of phenolics in plant tissues, and free phenolics are dominant in most plants.

### 2.2. Identification and Quantification of Phenolics in Three Different Organs of L. japonica and L. macranthoides

Figure 1 shows HPLC chromatograms of the free phenolic fractions in three different organs of *L. japonica* and *L. macranthoides*. It is clear from these chromatograms that peaks of phenolics occurred widely in the free phenolic fractions of flowers (Figure 1; curves c and f) and leaves (curves d and g) in the two species, but were comparatively scarce in those of their stems (curves b and e).

To characterize and compare phenolic compositions in *L. japonica* and *L. macranthoides*, free phenolic fractions from flowers and leaves of both species were further analyzed using UPLC-DAD–QTOF-MS/MS. By comparing their MS data, including retention time (Rt), experimental *m*/*z*, molecular formula, error of the experimental *m*/*z*, and MS^2^ fragments, with available literature and the MassBank MS database (http://www.massbank.jp/en/database.html), 10 phenolics were successfully identified (for the HPLC profile of their authentic standards, see curve a in Figure 1) and could be divided into three groups, i.e., 3-caffeoylquinic acid (3-CQA) and its derivatives (4-CQA, 5-CQA, 3,4-dicaffeoylquinic acid (3,4-diCQA), 3,5-diCQA, and 4,5-diCQA), three flavonoids (luteoloside, lonicerin, and isoquercitrin), and caffeic acid (Table 2).

Contents of the 10 detected phenolic compounds in the three phenolics fractions from the three different organs of *L. japonica* and *L. macranthoides*, obtained by calculation based on their corresponding standard curves (data not shown), are shown in Table 3. Overall, it is clear that chlorogenic acid (CGA) and its five derivatives were not detected in esterified and insoluble-bound phenolic fractions and were only detected in free phenolic fractions among all three organs of both species. On the other hand, almost all 10 detected phenolics were found from all free fractions, with only four exceptions in the leaf (isoquercitrin) and stem (luteoloside, isoquercitrin, and lonicerin) of *L. japonica* but not *L. macranthoides*. Furthermore, in addition to chlorogenic acid and its five derivatives, the other four phenolic compounds were also rarely detected from non-free fractions, with the sums of free fractions in flower, leaf, and stem being 6.4 and 6.5, 3.5 and 10.7, and 7.9 and 5.1 times higher in *L. japonica* and *L. macranthoides*, respectively, than their corresponding total amounts of the two non-free fractions. The last and most important point is that sums of the 10 phenolics in the three fractions of the three organs were much higher in *L. macranthoides* than in *L. japonica*, with the only exception being insoluble-bound fractions, which were a bit lower or comparably equal in the flower (0.83 vs. 0.97 mg/g DW) and leaf (3.19 vs. 3.92 mg/g DW).

In flowers of the two species (Table 3, top), the same phenolic compositions were found in each of the three fractions, and all 10 phenolics were identified in free fractions. In esterified fractions, caffeic acid and two flavonoids (luteoloside and isoquercitrin) were detected, while only caffeic acid was found in insoluble-bound fractions. Furthermore, phenolic contents were the highest in both free fractions, with the sum of *L. macranthoides* (86.55 mg/g DW) being much higher than that of *L. japonica* (52.40 mg/g DW).

In leaves of the two species (Table 3, middle), there existed few differences in phenolic compositions among the three fractions. Nine of the 10 phenolics were found in free fractions of both the two species, with the exception of isoquercitrin not being detected in *L. japonica*, and this compound, in addition to the three or two simultaneously found in both the two species, also occurred in either esterified or insoluble-bound fractions of *L. macranthoides*. Similarly, the highest phenolic contents existed in both free phenolic fractions, with the sum of *L. macranthoides* (115.97 mg/g DW) being also much higher than that of *L. japonica* (27.74 mg/g DW).

In stems of the two species (Table 3, bottom), differences in phenolic compositions among the three fractions were also obvious. All 10 phenolics were found from the free fraction of *L. macranthoides*, in which only caffeic acid and luteoloside were found from both non-free fractions, whereas three of the 10 phenolics, i.e., the only three flavonoids (luteoloside, isoquercitrin, and lonicerin), were not detected from the free fraction of *L. Japonica*, in which none and only caffeic acid were detected from its esterified and insoluble-bound fractions, respectively. Finally, once again, the highest phenolic contents also existed in both free phenolic fractions, with the sum of *L. macranthoides* (21.78 mg/g DW) being much higher than that of *L. japonica* (4.67 mg/g DW).

### 2.3. Differences in Antioxidant Activities of the Three Phenolic Fractions in L. japonica and L. macranthoides as Well as the 10 Identified Phenolics

For decades, synthetic antioxidants with characteristics of low cost and bland flavor were used as chemicals for food preservation by inhibiting oxidation processes. However, the demand for a “clean label” by consumers attracted great concern from researchers and necessitated the search for effective antioxidants from natural sources such as fruits, vegetables, and herbs that are beneficial for health promotion [28]. Three assays with different mechanisms were, thus, adopted to assess antioxidant capacities of the three phenolic fractions of total phenols and total flavonoids in the three organs of *L. japonica* and *L. macranthoides*. Two assays (i.e., DPPH and ABTS) were used for the assessment of antioxidant capacities of individual phenolics (0.2 mg/mL), including the 10 phenolics identified above and two more related dicaffeoylquinic acids (i.e., 1,3-CQA and 1,5-CQA). Specifically, ABTS is superior to DPPH when applied to samples containing hydrophilic, lipophilic, and highly pigmented antioxidant compounds [29].

As listed in Table 4, it is clear that activities in the DPPH•, ABTS^+^•, and oxygen radical absorbance capacity (ORAC) assays of the free phenolics maximized in the flower of *L. japonica* and the leaf of *L. macranthoides*, while the three highest activities of esterified phenolics were found in the flowers of both species, and those of the insoluble-bound phenolics were found in the leaf of *L. japonica* and the stem of *L. macranthoides*. Significantly stronger activities (see data on averages in Table 4) in all three antioxidant assays were observed in the free phenolics than in the esterified and insoluble-bound phenolics among all three organs (see data on averages in Table 4). Moreover, while esterified phenolics showed apparently higher antioxidant activities than insoluble-bound phenolics in *L. macranthoides*, they exhibited comparable activities in *L. japonica* in all the three assays (Table 4). Finally, in addition to the three activities of insoluble-bound phenolics that were similar in *L. japonica* and *L. macranthoides*, those of free and esterified phenolics in all three organs (Table 4) and total activities in each of the three organs (Table 4) were much stronger in *L. macranthoides* than in *L. japonica*. All the antioxidant results demonstrated that, among the three fractions, free phenolics were the strongest in antioxidant activities in all three organs, and *L. macranthoides* possessed superior antioxidant capacities to *L. japonica* (Table 4), which is consistent with the results of contents of total phenols and total flavonoids (Table 1).

For antioxidant activities of the 12 individual phenolics, as shown in Figure 2, obvious differences were found between data determined by DPPH and ABTS assays. Nevertheless, caffeic acid exhibited the best activities in both assays, with that in the former nearly two times higher than that in the later assays (0.37 vs. 0.20 mg Trolox/mL, respectively), stating clearly that it possessed the highest efficiency for scavenging free radicals within the 12 phenolics. In addition, three flavonoids including luteoloside, lonicerin, and isoquercitrin also showed much higher activities in DPPH assay than those in ABTS assay (0.28 vs. 0.11 mg Trolox/mL on average). On the other hand, both ABTS^+^• and DPPH• held similar sensitivities to the five dicaffeoylquinic acids (di-CQAs, 0.19 vs. 0.22) and three monocaffeoylquinic acids (mono-CQAs, 0.10 vs. 0.11). On average, di-CQAs possessed better activities than mono-CQAs in terms of scavenging both free radicals, indicating that more caffeic acid groups brought about better antioxidant activities. It is worth noting that the antioxidant activities of all 12 phenolics were quite similar, with maximum differences of only 4.26- and 2.11-fold in DPPH and ABTS assays, respectively.

Based on this principle that different phenolics possessed different antioxidant capacities, we determined antioxidant capacities of the 12 individual phenolics and multiplied each of them by the corresponding percentage of individual phenolic compounds in total content of all the phenolics in the free phenolic fraction (Table 5). Due to the difference in the content of different phenolics being far greater than the difference in their antioxidant activities (with the maximum differences being 113.02-fold and 4.26-fold, respectively), stronger antioxidant contributions were observed in phenolics with higher contents, such as 3-CQA and 3,5-diCQA (see Table 5).

Hsu et al. [5] compared DPPH• and ABTS^+^• scavenging activities in flowers of *L. japonica* var. *sempervillosa* and found that total phenols and total flavonoids, along with their better antioxidant activities, were higher in fresh flowers than those in dry flowers, indicating that the drying process led to decreases in antioxidant activities, as well as phenolic content. Earlier, Seo et al. [30] reported that DPPH• and ABTS^+^• scavenging activities of *L. japonica* were the strongest in the leaf, followed by the flower and stem, which is inconsistent with our results shown in Table 4; this might be caused by the different harvest seasons, as recently reported by Yang et al. [31], whereby *Acer truncatum* leaves harvested in different seasons possessed seasonal dynamics of constitutive levels of phenolic components, along with alterations of antioxidant capacities. Hu et al. [1] investigated antioxidant activities of flowers from *L. macranthoides* with four in vitro methods for various extracts (water extract, petroleum ether, ethyl acetate, and *n*-butanol fractions) and found that the best DPPH• scavenging activity and reducing power existed in the ethyl acetate fraction, and the *n*-butanol fraction possessed the best ABTS^+^• and superoxide anion scavenging activities. Up to date, no research was conducted on esterified and insoluble-bound phenolics of the two *Lonicera* species.

### 2.4. Differences in Protective Activities to DNA Damage of the Three Phenolic Fractions in L. japonica and L. macranthoides

DNA damage caused by oxidant by-products of normal metabolism (e.g., mitochondrial electron transport) is one of the major causes of aging and consequent degenerative diseases [32]. Under various stresses that usually produce excessive reactive oxygen species (ROS), supercoiled plasmid DNA might be converted to an open circular form followed by linear DNA via single- or double-strand breaks, and these conversions can be used as an indicator of DNA damage [33]. Due to the higher electrophoretic mobility of supercoiled DNA than that of its open circular counterpart, two separate DNA bands can be observed [34]. As shown in Figure 3A, all samples exhibited protective activities, in a dose-dependent manner, against oxidative damage to DNA caused by 2,2′-azobis(2-amidinopropane) dihydrochloride (AAPH). From the half maximal inhibitory concentration (IC_50_) values of inhibition to damage of supercoiled DNA shown in Figure 3B, it becomes obvious that the best protective activity against oxidative DNA damage of free phenolics was observed in the flower of *L. japonica* (0.13 mg/mL) and the leaf of *L. macranthoides* (0.07), and the best protective activities for esterified phenolics and insoluble-bound phenolics were in the flower (0.54) and leaf (0.48) of *L. japonica* and in the leaf (0.36) and stem (0.96) of *L. macranthoides*, respectively. It is worth emphasizing that free phenolics still possessed the best protective activities, followed by esterified and insoluble-bound phenolic fractions, and *L. macranthoides* also still exhibited better protective effects against damage to supercoiled DNA than *L. japonica* did, suggesting that higher levels of phenolics in certain organs of the two species, as well as in *L. macranthoides* than *L. japonica*, possessed not only stronger antioxidant activities as described above, but also better inhibitory effects against damage to supercoiled DNA.

Albishi et al. [35] reported that free, esterified, and insoluble-bound phenolic fractions of onion and potato peels possessed abundant polyphenols, which were used as free-radical scavengers and provided protection against DNA supercoiled strand scission induced by reactive oxygen species (ROS). In another study, Alshikh et al. [27] found that phenolic compounds of lentil in free, esterified, and insoluble-bound fractions were all effective in inhibiting DNA damage. Due to the scant information of DNA protective activities of *L. japonica* and *L. macranthoides*, our results, which also support the conclusions of the above literature, provide further evidence of inhibitive activities to DNA damage in these two species.

### 2.5. Correlation Analysis

Numerous studies showed that there exist significantly linear correlations between the content of phenolics and antioxidant capacity in various medicinal and edible plants. Table 6 shows comprehensive data obtained with a Pearson’s correlation coefficient analysis between several types of parameters.

In the free phenolic fractions (Table 6, top), there exists an extremely significant correlation between total phenols and total flavonoids, indicating that total flavonoids were the major contributor to total phenols. Correlations among three antioxidant capacities (DPPH, ABTS, and ORAC) were also extremely significant, while inhibition to DNA damage was negatively significantly correlated only with DPPH, indicating that radicals scavenged by DPPH contribute more to DNA damage. As to correlations among contents of individual phenolics, extremely significant correlations exist only among total quantified phenolics, 3-CQA, and 4,5-diCQA, and between 4-CQA and 5-CQA, whereas significant correlations exist between total quantified phenolics and both 4- and 5-CQAs, between luteoloside and lonicerin, between isoquercitrin and both 3,4- and 4,5-diCQAs, between 4,5-diCQA and both 4- and 5-CQAs, as well as both 3,4- and 3,5-diCQAs, and between 3-CQA and both 4- and 5-CQAs.

In the free phenolic fractions (Table 6, top), extremely significant correlations exist among total phenols, total flavonoids, and three antioxidant capacities, indicating that total phenols with flavonoids as the main components were responsible for the antioxidant capacities of this phenolic fraction, whereas negative correlations between inhibition to DNA damage and total phenols, as well as total flavonoids, were not significant, suggesting that compositions other than flavonoids such as 3,4-diCQA (see below) might play a role in damaging DNA. Total quantified phenolics, 3-CQA, and 4,5-diCQA (all extremely significantly) and 4- and 5-CQAs (only significantly) correlated with both total phenols and total flavonoids, with the exception that isoquercitrin was significantly correlated only with total phenols. On the other hand, total quantified phenolics and two individuals (3- and 4,5-CQAs) extremely significantly correlated with all three antioxidant capacities (DPPH, ABTS, and ORAC), while 4- and 5-CQAs showed the same level of correlations only with ORAC. Significant correlations exist between DPPH and isoquercitrin, as well as 3,4- and 3,5-CQAs, and between ABTS and 4- and 5-CQAs, whereas inhibition to DNA damage exhibits a negatively significant correlation only with 3,4-diCQA, confirming the above suggestion.

Roughly similar patterns of correlations to those in the free phenolic fractions exist in the esterified phenolic fractions (Table 6, middle). What is worth noting here is that extremely significant correlations exist between caffeic acid and both total quantified phenolics and total phenols, as well as total flavonoids, between total quantified phenolics and all three antioxidant capacities (DPPH, ABTS, and ORAC), and between caffeic acid and two of three antioxidant capacities (DPPH and ABTS), while luteoloside was negatively extremely significantly correlated with inhibition to DNA damage, suggesting that total quantified phenolics, especially caffeic acid, contributed more to the scavenging capabilities, and luteoloside was the main factor reducing DNA damage.

Finally, in the insoluble-bound phenolic fractions (Table 6, bottom), two aspects worth highlighting are that extremely significant correlations exist between luteoloside and total quantified phenolics, and between caffeic acid and both total phenols and total flavonoids, as well as all three radical scavenging activities (DPPH, ABTS, and ORAC), while inhibition to DNA damage showed negatively significant correlations with all three antioxidant activities (DPPH, ABTS, and ORAC), total phenols, total flavonoids, and caffeic acid. All these suggest that this phenolic fraction contains certain compositions (e.g., caffeic acid and/or luteoloside) that contributed to inhibition to DNA damage, as well as to antioxidant activities.

## 3. Materials and Methods

### 3.1. Chemical Reagents and Plant Materials

2,2-Diphenyl-1-picrylhydrazyl (DPPH), 2,2′-azino-bis(3-ethylbenzthiazoline-6-sulfonate) (ABTS), 2,2′-azobis(2-amidinopropane) dihydrochloride (AAPH), Folin–Ciocalteu, agarose, and 6-hydroxy-2,5,7,8-tetramethylchromane-2-carboxylic acid (Trolox) were purchased from Sigma-Aldrich Chemical (St. Louis, MO, USA). The pBR 322 plasmid DNA was purchased from Thermo Fisher Scientific Inc. (Waltham, MA, USA). Tris-acetate-ethylenediaminetetraacetic acid (EDTA) (TAE) buffer, GelRed^TM^ nucleic acid stain, xylene cyanol, glycerol, and bromophenol blue were purchased from BioDee Biotechnology Co. Ltd. (Beijing, China). Authentic phenolic standards, e.g., 3-*O*-caffeoylquinic acid (3-CQA), 4-*O*-caffeoylquinic acid (4-CQA), 5-*O*-caffeoylquinic acid (5-CQA), 3,4-di-*O*-caffeoylquinic acid (3,4-diCQA), 3,5-di-*O*-caffeoylquinic acid (3,5-diCQA), 4,5-di-*O*-caffeoylquinic acid (4,5-diCQA), 1,3-di-*O*-caffeoylquinic acid (1,3-diCQA), 1,5-di-*O*-caffeoylquinic acid (1,5-diCQA), caffeic acid, luteoloside, lonicerin, and isoquercitrin, were purchased from Lyle Wormwood Biological Technology Co. (Luoyang, Henan province, China). HPLC-grade methanol and formic acid were purchased from Tedia Co. (Fairfield, OH, USA). All other chemicals and reagents used in the experiments were of analytical grade.

Air-dried flowers, leaves, and stems of *Lonicera japonica* Thunb. and *L. macranthoides* Hand.-Mazz. were purchased from Longhui county, Hunan province, and Pingyi county, Shandong province of China, respectively. After being authenticated by Professor Zhonghua Liu at Beijing Forestry University, China, plant materials further dried at 60 °C overnight in an oven were ground into fine powder followed by passing through a 40-mesh sieve; they were stored at −20 °C in polyethylene bags for further use.

### 3.2. Extraction of Phenolic Compounds

Free, esterified, and insoluble-bound phenolics from flowers, leaves, and stems of *L. japonica* and *L. macranthoides* were prepared using the methods reported in the literature [19,25] with slight modifications. In brief, 1.0000 g of powder was ultrasonicated with 15 mL of aqueous methanol (70%, *v*/*v*) for 30 min at room temperature. The mixture was then centrifuged at 6000 rpm for 15 min, the residue was treated twice with the same solvent, and the three supernatants were combined. After evaporation at 50 °C through a rotary evaporator to remove the methanol, the water phase was acidified (pH 2.0) with 6 M HCl, followed by extracting three times with ethyl acetate (1:1, *v*/*v*). After combination, the dried extracts achieved by evaporating under vacuum at 50 °C were dissolved in 30 mL of 70% aqueous methanol to obtain the free phenolic fraction. The free phenolic fractions were filtrated through a 0.22-μm filter and stored at 4 °C for subsequent analyses.

To obtain the esterified phenolic fraction, 30 mL of 4 M NaOH was added to the upper aqueous phase containing esterified phenolics and was subsequently hydrolyzed for 4 h at room temperature. After the acidification treatment (pH 2.0) with 6 M HCl, the resultant hydrolysate was subsequently extracted using ethyl acetate three times. After combining the ethyl acetate phase, the dried extracts achieved by evaporating under vacuum at 50 °C were subsequently dissolved in 30 mL of 70% aqueous methanol to obtain phenolics released from their esterified form. The esterified phenolic fractions were filtrated through a 0.22-μm filter and stored at 4 °C for subsequent analyses.

To obtain insoluble-bound phenolic fraction simultaneously, the solid residues were hydrolyzed with 20 mL of 4 M NaOH for 4 h at room temperature. After acidification (pH 2.0) with 6 M HCl and centrifugation at 6000 rpm for 15 min, the supernatant was subsequently extracted using ethyl acetate three times. Before dissolving in 30 mL of 70% aqueous methanol, the combined extracts containing phenolics released from insoluble-bound form were evaporated to dryness under the same evaporation conditions as before. The insoluble-bound phenolic fractions were also filtrated through a 0.22-μm filter and stored at 4 °C for subsequent analyses.

### 3.3. Determination of Phenolic Compounds

Total phenols of each fraction were determined according to the Folin–Ciocalteu method [36] with slight modifications. Briefly, 20 μL of standard (i.e., 0–400 mg/L gallic acid), sample, or blank (Milli-Q water) was added to 40 μL of 50% Folin–Ciocalteu solution in corresponding wells of a 96-well microplate. Subsequently, 140 μL of Na_2_CO_3_ solution (700 mM) was added and the microplate was shaken for 6 min at 200 rpm. Absorbance of the mixture was measured at 765 nm using a microplate reader (Bio-Rad xMark^TM^ Microplate Absorbance Spectrophotometer, Hercules, CA, USA) after incubation in the dark at 40 °C for 30 min. Results were expressed as μmol gallic acid equivalent per gram of dry weight (μmol GAE/g DW).

Total flavonoids of each fraction were determined by an aluminum chloride colorimetric assay [36] with slight modifications. In brief, 120 μL of standard (i.e., 10–100 mg/L rutin), sample, or blank was mixed with 8 μL of 50 mg/mL NaNO_2_ in corresponding wells of a 96-well microplate. Six minutes later, 8 μL of 100 mg/mL AlCl_3_ was pipetted into each well and was allowed to stand for another 5 min at room temperature before 100 μL of 40 mg/mL NaOH was added. Subsequently, the microplate with the well-mixed mixture was covered and incubated in the dark at room temperature for 30 min. Absorbance of the mixture was then measured at 410 nm using the microplate reader, and results were expressed as μmol rutin equivalent per gram of dry weight (μmol RE/g DW).

### 3.4. Evaluation of Radical Scavenging Activities

DPPH• scavenging activity of each fraction was evaluated using the method reported in the literature [37] with modifications. Briefly, 40 μL of fresh DPPH solution (1.0 mM) was mixed well with 10 μL of standard (i.e., 0–400 mg/L Trolox), sample, or blank in corresponding wells of a 96-well microplate before 190 μL of methanol was added. After shaking at 200 rpm for 6 min with an orbital shaker, the microplate was incubated for 30 min in the dark at room temperature and absorbance at 517 nm was measured using the microplate reader. Results were expressed as μmol Trolox equivalents per gram of dry weight (μmol Trolox/g DW).

ABTS^+^• scavenging capacity of each fraction was evaluated according to the method reported by Alanon et al. [37] with modifications. ABTS^+^• stock solution (7 mM ABTS and 2.4 mM K_2_S_2_O_8_ in equivalents) was prepared and stood for 12–16 h in the dark at room temperature, and ABTS^+^• working solution was generated by diluting the stock solution with methanol at a ratio of 1:48 to an absorbance of 0.70 ± 0.02 measured at 734 nm using the microplate reader. Subsequently, 200 μL of ABTS^+^• working solution was mixed well with 5 μL of standard (i.e., 0–800 mg/L Trolox), sample, or blank in corresponding wells of a 96-well microplate. Absorbance at 734 nm was measured after incubating at 30 °C for 5 min in the dark. Results were expressed as μmol Trolox equivalents per gram of dry weight (μmol Trolox/g DW).

Oxygen radical absorbance capacity (ORAC) of each phenolic fraction was evaluated using the procedure reported by Sun et al. [38] with modifications. All reagents were prepared in a 75-mM phosphate-buffered solution (PBS, pH 7.4) and the whole process was protected from direct light. In brief, 75 μL of fluorescein solution (0.20 μM) and 25 μL of standard (i.e., 5–50 μM Trolox), sample, or blank (75 mM PBS) were mixed well in corresponding wells of a 96-well microplate and incubated at 37 °C in the dark for 15 min. Immediately before measurement every 1.5 min for 75 min with an excitation at 530 nm and emission at 485 nm, 100 μL of 37 °C prewarmed AAPH was quickly pipetted into each well. The ORAC values were calculated using the net areas under the curve (AUC) of samples and standards subtracted by the AUC of the blank, and results were expressed as μmol Trolox equivalents per gram of dry weight (μmol Trolox/g DW).

### 3.5. Examination of Inhibition to DNA Damage

Inhibition of AAPH-induced damage to supercoiled DNA by each of the three phenolic fractions was examined according to the method of Albishi et al. [35] with a slight modification. All reagents were dissolved in a 75 mM phosphate-buffered solution (PBS, pH 7.4). Briefly, 4 μL of supercoiled plasmid DNA solution (pBR 322 from *Escherichia coli* RRI, 50 μg/mL), 4 μL of each sample at different concentrations, and 4 μL of 9 mM AAPH were added sequentially to 4 μL of PBS in an Eppendorf tube. After centrifugation and incubation at 37 °C for 1 h, 4 μL of loading dye consisting of 0.25% bromophenol blue, 0.25% xylene cyanol, and 50% glycerol in distilled water was added to the reaction mixture. A 0.7% (*w*/*v*) agarose gel was prepared in 50× TAE buffer containing GelRed^TM^ nucleic acid stain (10 μL/100 mL of gel). Each sample was loaded, and gel electrophoresis was run (60 V) for 1 h at room temperature in the TAE buffer. DNA bands were then imaged, and band intensities were analyzed using Azurespot software (Azure Biosystems Inc, Dublin, CA, USA). Results of inhibition to supercoiled DNA damage were expressed as IC_50_ values (mg/mL), i.e., the concentration of a sample that causes 50% inhibition.

### 3.6. UPLC-DAD–QTOF-MS/MS Analysis

Ultra-performance liquid chromatography with a diode array detector, alongside tandem mass spectrometry coupled with a quadrupole time-of-flight mass spectrometer (UPLC–DAD–MS/MS) was used to determine phenolic compositions of the free phenolic fractions from flowers and leaves of both *L. japonica* and *L. macranthoides*, while phenolic compositions in the other two phenolic fractions were determined by comparing retention times of the integrated peaks of the standards. Both negative- and positive-ion modes were operated during the MS/MS measurements. Phenolics were separated by a reversed-phase C_18_ column (Diamonsil 250 × 4.6 mm inner diameter (i.d.), 5 μm, China) at 30 °C with an injection volume of 10 μL, and the mobile phase was composed of A (1% formic acid in water) and B (methanol) at a flow rate of 1.0 mL/min. A linear gradient program was set as follows: 0–4 min, 12–18% B; 4–30 min, 18–28% B; 30–75 min, 28–35% B; 75–120 min, 35–38% B. Phenolics were detected at 330 nm, and their contents were quantified by HPLC chromatograms monitored at 330 nm through comparing integrated peak areas of the standards with those of corresponding phenolic compounds to be identified. Results were expressed as mg per gram of dry weight (mg/g DW).

### 3.7. Statistical Analysis

Data were expressed as means ± standard deviation. A one-way ANOVA analysis using SPSS 16.0 (SPSS Inc., Chicago, IL, USA) was used to determine statistical significance among variables with a significance level set at *p* < 0.05. The Duncan test was used for the expression of significant differences, with distinct letters next to data values in tables and figures. Pearson’s correlation coefficient analysis was performed using OriginPro 2017 (OriginLab Corp., Northampton, MA, USA) to explore the relationship among these tested variables.

## 4. Conclusions

This study investigated and compared antioxidant activities of free, esterified, and insoluble-bound phenolic fractions in *L. japonica* and *L. macranthoides*. The results show that extracts from three different organs of the two species possessed considerable antioxidant activity against different radicals, including DPPH•, ABTS^+^•, and peroxyl radicals. Compared with the activities of esterified and insoluble-bound fractions of phenolics, significantly stronger antioxidant activities of the free phenolics were obtained in each of the three organs, which is in accordance with their contents of total phenols and total flavonoids. Additionally, in identical fractions between the same organs of the two species, *L. macranthoides* exhibited superior antioxidant activities and better protective activity against oxidative stress, as well as contents of the three phenolic fractions, than *L. japonica* did in most cases, and fractions with higher total phenols and total flavonoids were also more effective in protecting DNA from radical-induced oxidative damage. Ten phenolics consisting of 3-caffeoylquinic acid (3-CQA) and its five derivatives (4-CQA, 5-CQA, 3,4-diCQA, 3,5-diCQA, 4,5-diCQA), caffeic acid, and three flavonoids (luteoloside, lonicerin, and isoquercitrin) were further identified and quantified by UPLC-DAD–QTOF-MS/MS. Chlorogenic acid and its five derivatives were found only in free fractions, closely correlated with antioxidant activities, while caffeic acid was the major contributor to the antioxidant activities of esterified and insoluble-bound phenolic fractions. It is, thus, concluded that, like *L. japonica*, *L. macranthoides*, which was underestimated since being separately listed by the 2010 edition of the Chinese Pharmacopoeia, is also a good (and better) herbal medicine.

## Figures and Tables

**Figure 1 molecules-24-00970-f001:**
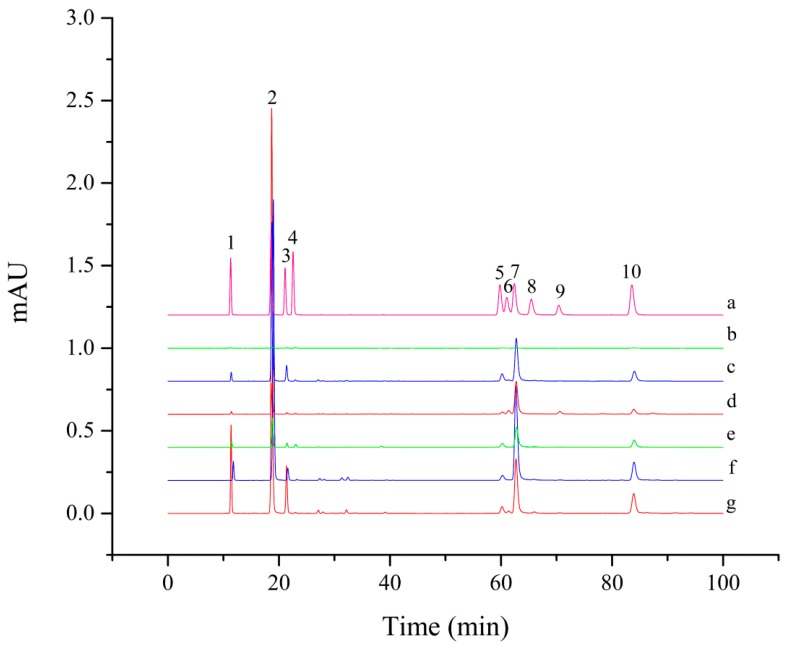
HPLC chromatograms of 10 phenolic standards and free phenolic fractions in three different organs of *Lonicera japonica* and *L. macranthoides.* (a) Ten phenolics standards; (b–d), free phenolic fractions from stem, flower, and leaf of *L. japonica*, respectively; (e–g), free phenolic fractions from stem, flower, and leaf of *L. macranthoides*, respectively. Peaks are numbered from left to right with the following sequence: (1) 5-caffeoylquinic acid (5-CQA); (2) 3-CQA; (3) 4-CQA; (4) caffeic acid; (5) 3,4-dicaffeoylquinic acid (3,4-diCQA); (6) luteoloside; (7) 3,5-diCQA; (8) isoquercitrin; (9) lonicerin; (10) 4,5-diCQA. The same colors indicate identical organs.

**Figure 2 molecules-24-00970-f002:**
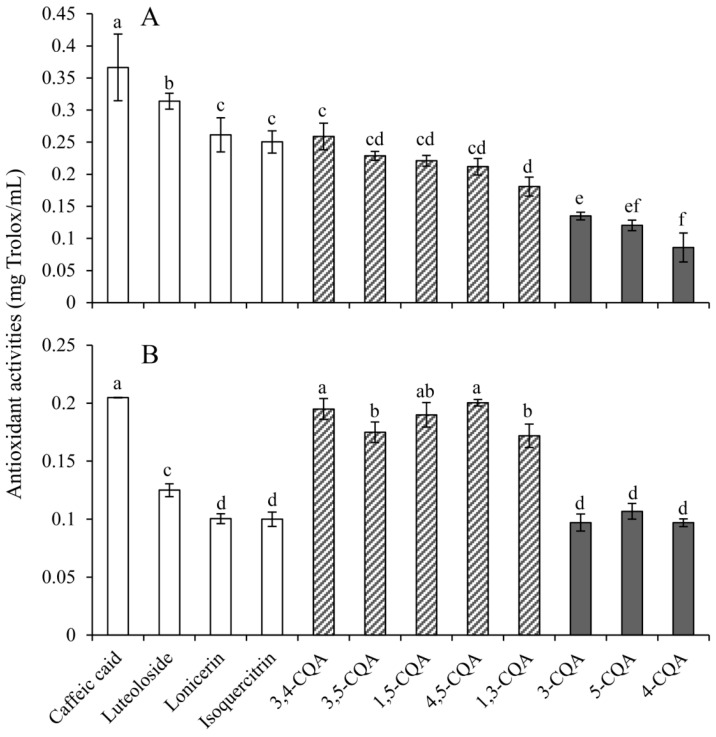
Antioxidant activities (mg Trolox/mL) of the 12 individual phenolics by 2,2-diphenyl-1-picrylhydrazyl (DPPH) (**A**) and 2,2′-azino-bis(3-ethylbenzthiazoline-6-sulfonate) (ABTS) (**B**) assays. Phenolics are divided into three groups from left to right with the following sequence: non-chlorogenic acid and three flavonoids (luteoloside, lonicerin, and isoquercitrin), five di-CQAs (3,4-diCQA, 3,5-diCQA, 1,5-diCQA, 4,5-diCQA, and 1,3-diCQA), and three mono-CQAs (3-CQA, 5-CQA and 4-CQA). Data are given as means ± SD (*n* = 3). Different lowercase letters (a–f) indicate significant differences (*p* < 0.05).

**Figure 3 molecules-24-00970-f003:**
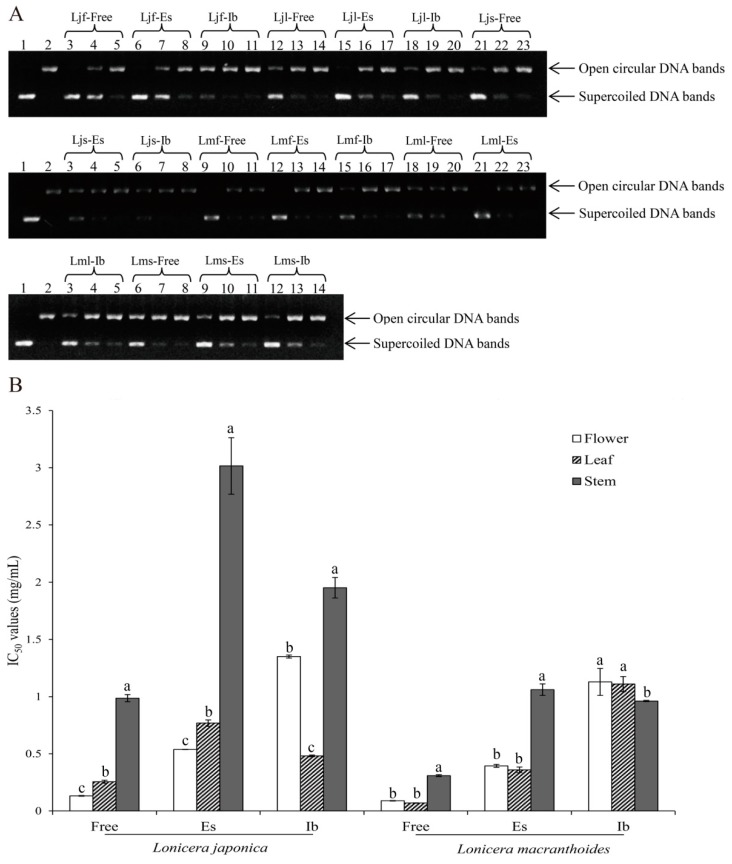
Protective activities to DNA damage of three phenolic fractions of total phenols and total flavonoids from three organs of *Lonicera japonica* and *L. macranthoides*. (**A**) Agarose gel electrophoretogram of DNA treated with phosphate-buffered saline (PBS; lane 1), 2,2′-azobis(2-amidinopropane) dihydrochloride (AAPH; lane 2), and AAPH + samples at different concentrations (lanes 3–23). (**B**) Half maximal inhibitory concentration (IC_50_) of protective activity to AAPH-induced DNA damage (mg/mL). Ljf, Lmf, Ljl, Lml, Ljs, and Lms represent flowers, leaves, and stems, and Free, Es, and Ib represent free, esterified, and insoluble-bound phenolic fractions of *L. japonica* and *L. macranthoides*, respectively. Different lowercase letters (a–c) indicate significant differences (*p* < 0.05).

**Table 1 molecules-24-00970-t001:** Total phenols and total flavonoids of free, esterified, and insoluble-bound (I-B) phenolics.

Species	Organs	Free	Esterified	I-B	Sum
**Total Phenols (μmol Gallic Acid Equivalent per Gram of Dry Weight (μmol GAE/g DW))**					
*Lonicera japonica*	Flower	150.44 ± 1.43 ^Ca^	38.80 ± 0.90 ^Cb^	6.16 ± 0.21 ^Ec^	195.40
	Leaf	98.79 ± 0.98 ^Da^	21.67 ± 0.30 ^Dc^	29.97 ± 0.27 ^Ab^	150.42
	Stem	15.78 ± 0.34 ^Ea^	1.84 ± 0.21 ^Fc^	6.09 ± 0.24 ^Eb^	23.71
	Average	88.34	20.77	14.07	
*Lonicera macranthoides*	Flower	265.13 ± 2.28 ^Ba^	77.04 ± 0.51 ^Ab^	11.07 ± 0.26 ^Dc^	353.25
	Leaf	365.83 ± 6.32 ^Aa^	59.77 ± 1.04 ^Bb^	13.54 ± 0.17 ^Cc^	439.13
	Stem	97.08 ± 1.21 ^Da^	12.16 ± 0.52 ^Ec^	16.10 ± 0.59 ^Bb^	125.34
	Average	242.68	49.66	13.57	
**Total Flavonoids (μmol Rutin Equivalent per Gram of Dry Weight (μmol RE/g DW))**					
*Lonicera japonica*	Flower	60.64 ± 0.42 ^Ca^	16.39 ± 0.57 ^Cb^	3.41 ± 0.02 ^Ec^	80.44
	Leaf	55.09 ± 0.55 ^Da^	11.63 ± 0.33 ^Dc^	12.84 ± 0.08 ^Ab^	79.56
	Stem	7.87 ± 0.05 ^Fa^	0.93 ± 0.00 ^Fc^	2.83 ± 0.05 ^Fb^	11.62
	Average	41.20	9.65	6.36	
*Lonicera macranthoides*	Flower	106.46 ± 0.30 ^Ba^	33.96 ± 0.35 ^Ab^	4.87 ± 0.03 ^Dc^	145.29
	Leaf	166.59 ± 1.87 ^Aa^	25.88 ± 0.31 ^Bb^	6.45 ± 0.27 ^Cc^	198.91
	Stem	45.96 ± 0.88 ^Ea^	5.17 ± 0.05 ^Ec^	7.00 ± 0.11 ^Bb^	58.13
	Average	106.34	21.67	6.11	

Different upper- and lowercase letters within a column and row, respectively, indicate significant differences (*p* < 0.05); Free, free phenolics; Esterified, esterified phenolics; I-B, insoluble-bound phenolics.

**Table 2 molecules-24-00970-t002:** Identification of phenolics by ultra-performance liquid chromatography with a diode array detector, alongside tandem mass spectrometry coupled with a quadrupole time-of-flight mass spectrometer (UPLC-DAD–QTOF-MS/MS). Rt—retention time; CQA—caffeoylquinic acid; diCQA—dicaffeoylquinic acid.

No.	Rt (min)	Error (ppm)	Measured Value (*m*/*z*)	MS/MS Fragments *m*/*z*	Molecular Formula	Identified Phenols
1	10.10	−2.3	353.09	191.05 [M − (caffeic acid − H_2_O) − H]^−^ 179.03 [M − (quinic acid − H_2_O) − H]^−^ 135.05 [M − (quinic acid − H_2_O) − CO_2_ − H]^−^	C_16_H_17_O_9_	5-CQA
2	17.73	1.1	353.09	191.06 [M − (caffeic acid − H_2_O) − H]^−^	C_16_H_17_O_9_	3-CQA
3	20.60	0.8	353.09	173.04 [M − (caffeic acid − H_2_O) − H_2_O − H]^−^ 179.04 [M − (quinic acid − H_2_O) − H]^−^ 191.06 [M − (caffeic acid − H_2_O) − H]^−^ 135.04 [M − (quinic acid − H_2_O) − CO_2_ − H]^−^	C_16_H_17_O_9_	4-CQA
4	22.77	10.6	179.03	135.05 [M − CO_2_ − H]^−^	C_9_H_7_O_4_	Caffeic acid
5	61.17	2.9	515.12	179.04 [M − (caffeic acid-H_2_O) − (quinic acid − H_2_O) − H]^−^ 173.05 [M − 2(caffeic acid − H_2_O) − H_2_O − H]^−^ 353.09 [M − (caffeic acid − H_2_O) − H]^−^ 191.06 [M − 2(caffeic acid − H_2_O) − H]^−^	C_25_H_23_O_12_	3,4-diCQA
6	62.88	3.1	447.09	285.04 [M − (Glu − H_2_O) − H]^−^	C_21_H_19_O_11_	Luteoloside
7	64.46	−2.7	515.12	191.06 [M − 2(caffeic acid − H_2_O) − H]^−^ 353.09 [M − (caffeic acid− H_2_O) − H]^−^ 179.03 [M − (caffeic acid − H_2_O) − (quinic acid − H_2_O) − H]^−^ 135.04 [M − (caffeic acid − H_2_O) − (quinic acid − H_2_O) − CO_2_ − H]^−^	C_25_H_23_O_12_	3,5-diCQA
8	68.80	4.5	463.09	301.032 [M − (Glu − H_2_O) − H]^−^	C_21_H_19_O_12_	Isoquercitrin
9	71.40	0.7	593.15	285.04 [M − (rutinose − H_2_O) − H]^−^	C_27_H_29_O_15_	Lonicerin
10	85.76	−4.1	515.12	173.05 [M − 2(caffeic acid − H_2_O) − H_2_O − H]^−^ 353.09 [M − (caffeic acid − H_2_O) − H]^−^ 179.03 [M − (caffeic acid − H_2_O) − (quinic acid − H_2_O) − H]^−^ 191.0551 [M − 2(caffeic acid − H_2_O) − H]^−^	C_25_H_23_O_12_	4,5-diCQA

**Table 3 molecules-24-00970-t003:** Contents of major phenolics (mg/g DW).

No.	Phenolics	*Lonicera japonica*			*Lonicera macranthoides*		
		Free	Esterified	I-B	Free	Esterified	I-B
**Flower**							
1	5-CQA	1.61 ± 0.05 ^c^	-	-	2.80 ± 0.12 ^b^	-	-
2	3-CQA	23.49 ± 0.69 ^c^	-	-	42.14 ± 1.42 ^b^	-	-
3	4-CQA	3.06 ± 0.08 ^b^	-	-	2.50 ± 0.07 ^c^	-	-
4	Caffeic acid	0.54 ± 0.01 ^i^	4.96 ± 0.11 ^b^	0.97 ± 0.02 ^g^	0.60 ± 0.10 ^i^	10.42 ± 0.16 ^a^	0.83 ± 0.01 ^h^
5	3,4-diCQA	3.58 ± 0.04 ^a^		-	3.02 ± 0.12 ^c^	-	-
6	Luteoloside	1.54 ± 0.02 ^c^	1.24 ± 0.04 ^d^	-	1.16 ± 0.08 ^defg^	1.15 ± 0.00 ^defg^	-
7	3,5-diCQA	12.81 ± 0.57 ^c^	-	-	26.19 ± 1.82 ^a^	-	-
8	Isoquercitrin	1.19 ± 0.02 ^c^	1.00 ± 0.01 ^e^	-	1.28 ± 0.04 ^b^	0.99 ± 0.01 ^e^	-
9	Lonicerin	0.87 ± 0.05 ^de^	-	-	0.95 ± 0.07 ^d^	-	-
10	4,5-diCQA	3.71 ± 0.21 ^c^	-	-	5.92 ± 0.53 ^b^	-	-
	Sum	52.40	7.20	0.97	86.55	12.56	0.83
**Leaf**							
1	5-CQA	0.85 ± 0.05 ^e^	-	-	11.73 ± 0.28 ^a^	-	-
2	3-CQA	5.39 ± 0.14 ^d^	-	-	63.29 ± 2.18 ^a^	-	-
3	4-CQA	0.81 ± 0.01 ^e^	-	-	8.82 ± 0.12 ^a^	-	-
4	Caffeic acid	0.50 ± 0.01 ^i^	1.28 ± 0.03 ^e^	2.68 ± 0.05 ^d^	0.56 ± 0.04 ^i^	4.72 ± 0.05 ^c^	1.15 ± 0.02 ^f^
5	3,4-diCQA	1.68 ± 0.03 ^e^	-	-	3.36 ± 0.03 ^b^	-	-
6	Luteoloside	2.65 ± 0.08 ^a^	1.18 ± 0.01 ^def^	1.24 ± 0.01 ^d^	2.04 ± 0.03 ^b^	1.21 ± 0.00 ^de^	1.10 ± 0.01 ^fg^
7	3,5-diCQA	11.10 ± 0.36 ^d^	-	-	16.25 ± 0.58 ^b^	-	-
8	Isoquercitrin	-	-	-	1.65 ± 0.04 ^a^	1.03 ± 0.01 ^e^	0.93 ± 0.01 ^f^
9	Lonicerin	2.50 ± 0.10 ^a^	1.51 ± 0.06 ^b^	-	1.07 ± 0.05 ^c^	0.73 ± 0.01 ^f^	-
10	4,5-diCQA	2.27 ± 0.09 ^d^	-	-	7.20 ± 0.15 ^a^	-	-
	Sum	27.74	3.98	3.92	115.97	7.69	3.19
**Stem**							
1	5-CQA	0.51 ± 0.00 ^f^	-	-	1.14 ± 0.04 ^d^	-	-
2	3-CQA	0.87 ± 0.02 ^e^	-	-	4.77 ± 0.24 ^d^	-	-
3	4-CQA	0.60 ± 0.00 ^f^	-	-	1.22 ± 0.02 ^d^	-	-
4	Caffeic acid	0.51 ± 0.00 ^i^	-	0.59 ± 0.01 ^i^	0.74 ± 0.01 ^h^	0.84 ± 0.01 ^h^	1.21 ± 0.01 ^ef^
5	3,4-diCQA	0.88 ± 0.01 ^f^	-	-	2.21 ± 0.15 ^d^	-	-
6	Luteoloside	-	-	-	1.08 ± 0.00 ^g^	1.13 ± 0.00 ^efg^	1.11 ± 0.01 ^fg^
7	3,5-diCQA	0.60 ± 0.01 ^f^	-	-	6.09 ± 0.66 ^e^	-	-
8	Isoquercitrin	-	-	-	1.09 ± 0.02 ^d^	-	-
9	Lonicerin	-	-	-	0.79 ± 0.04 ^ef^	-	-
10	4,5-diCQA	0.71 ± 0.02 ^e^	-	-	2.65 ± 0.29 ^d^	-	-
	Sum	4.67	-	0.59	21.78	1.97	2.32

Different lowercase letters in a row indicate significant differences (*p* < 0.05); Free, free phenol fractions; Esterified, esterified phenol fractions; I-B, insoluble-bound phenol fractions; - indicates phenolics not detected.

**Table 4 molecules-24-00970-t004:** 2,2-Diphenyl-1-picrylhydrazyl radical (DPPH•) and 2,2′-azino-bis(3-ethylbenzthiazoline-6-sulfonate) radical (ABTS^+^•) activities, and oxygen radical absorbance capacity (ORAC) of free, esterified, and insoluble-bound phenolic fractions (μmol Trolox/g DW).

Species	Organs	Free	Esterified	I-B	Sum
**DPPH**					
*Lonicera japonica*	Flower	178.60 ± 2.29 ^Ca^	29.91 ± 0.96 ^Cb^	10.58 ± 0.62 ^Dc^	219.08
	Leaf	98.25 ± 3.19 ^Da^	18.36 ± 0.36 ^Dc^	32.86 ± 0.66 ^Ab^	149.46
	Stem	11.18 ± 0.40 ^Ea^	1.07 ± 0.70 ^Fc^	7.89 ± 0.27 ^Eb^	20.14
	Average	96.01	16.45	17.11	
*Lonicera macranthoides*	Flower	279.14 ± 1.85 ^Ba^	88.07 ± 2.22 ^Ab^	12.22 ± 0.53 ^Cc^	379.44
	Leaf	293.08 ± 5.98 ^Aa^	53.98 ± 2.02 ^Bb^	12.02 ± 0.53 ^Dc^	359.07
	Stem	99.77 ± 7.70 ^Da^	8.82 ± 0.28 ^Eb^	15.22 ± 0.44 ^Bb^	123.81
	Average	224.00	50.29	13.15	
**ABTS**					
*Lonicera japonica*	Flower	132.96 ± 1.37 ^Ca^	41.04 ± 0.37 ^Cb^	11.70 ± 0.63 ^Dc^	185.70
	Leaf	113.67 ± 2.41 ^Da^	22.72 ± 0.36 ^Dc^	43.28 ± 1.19 ^Ab^	179.67
	Stem	23.67 ± 0.64 ^Ea^	2.21 ± 0.07 ^Fc^	10.55 ± 0.22 ^Eb^	36.43
	Average	90.10	21.99	21.84	
*Lonicera macranthoides*	Flower	238.96 ± 7.79 ^Ba^	95.80 ± 1.23 ^Ab^	19.92 ± 0.53 ^Cc^	354.68
	Leaf	329.08 ± 5.09 ^Aa^	77.33 ± 0.52 ^Bb^	20.21 ± 0.09 ^Cc^	426.61
	Stem	108.28 ± 1.40 ^Da^	16.39 ± 0.12 ^Ec^	22.54 ± 0.37 ^Bb^	147.20
	Average	225.44	63.17	20.76	
**ORAC**					
*Lonicera japonica*	Flower	1758.76 ± 29.04 ^Ca^	532.55 ± 5.87 ^Cb^	122.56 ± 16.85 ^Ec^	2413.87
	Leaf	1269.89 ± 77.93 ^Da^	331.75 ± 10.69 ^Dc^	453.83 ± 7.31 ^Ab^	2055.47
	Stem	284.88 ± 13.82 ^Ea^	28.30 ± 6.78 ^Fc^	97.68 ± 5.11 ^Fb^	410.86
	Average	1104.51	297.53	224.69	
*Lonicera macranthoides*	Flower	2910.83 ± 100.44 ^Ba^	1083.59 ± 2.65 ^Ab^	180.12 ± 12.08 ^Dc^	4174.54
	Leaf	4637.35 ± 137.41 ^Aa^	942.67 ± 34.71 ^Bb^	248.16 ± 1.62 ^Cc^	5828.19
	Stem	1133.83 ± 80.79 ^Da^	211.11 ± 9.11 ^Eb^	272.37 ± 17.40 ^Bb^	1617.30
	Average	2894.00	745.79	233.55	

Different upper- and lowercase letters within a column and row, respectively, indicate significant differences (*p* < 0.05); Free, free phenolics; Esterified, esterified phenolics; I-B, insoluble-bound phenolics.

**Table 5 molecules-24-00970-t005:** Contributions of free phenolics to total antioxidant activities in DPPH and ABTS assays.

Phenolics *	Contribution					
	Flower		Leaf		Stem	
	*L. japonica*	*L. macranthoides*	*L. japonica*	*L. macranthoides*	*L. japonica*	*L. macranthoides*
DPPH activity						
Caffeic acid (4) **	1.90 ± 0.05	1.26 ± 0.04	3.33 ± 0.09	0.88 ± 0.02	19.93 ± 0.56	6.26 ± 0.18
Luteoloside (6)	4.62 ± 0.04	2.10 ± 0.02	15.00 ± 0.12	2.77 ± 0.02	-	7.79 ± 0.06
Lonicerin (9)	2.17 ± 0.04	1.43 ± 0.03	11.78 ± 0.24	1.20 ± 0.02	-	4.72 ± 0.10
Isoquercitrin (8)	2.84 ± 0.04	1.85 ± 0.03	-	1.79 ± 0.02	-	6.26 ± 0.09
3,4-CQA (5)	8.84 ± 0.14	4.52 ± 0.07	7.82 ± 0.13	3.75 ± 0.06	24.52 ± 0.39	13.12 ± 0.21
3,5-CQA (7)	27.95 ± 0.17	34.60 ± 0.21	45.73 ± 0.28	16.02 ± 0.10	14.59 ± 0.09	31.98 ± 0.19
4,5-CQA (10)	7.49 ± 0.09	7.24 ± 0.09	8.66 ± 0.11	6.58 ± 0.08	16.10 ± 0.20	12.89 ± 0.16
3-CQA (2)	30.24 ± 0.27	32.84 ± 0.30	13.09 ± 0.12	36.81 ± 0.33	12.53 ± 0.11	14.79 ± 0.13
5-CQA (1)	1.85 ± 0.02	1.95 ± 0.03	1.84 ± 0.02	6.09 ± 0.08	6.51 ± 0.09	3.15 ± 0.04
4-CQA (3)	2.51 ± 0.13	1.24 ± 0.07	1.26 ± 0.07	3.27 ± 0.17	5.50 ± 0.29	2.40 ± 0.13
ABTS activity						
Caffeic acid (4)	1.06 ± 0.00	0.71 ± 0.00	1.86 ± 0.00	0.49 ± 0.00	11.13 ± 0.00	3.50 ± 0.00
Luteoloside (6)	1.84 ± 0.02	0.83 ± 0.01	5.97 ± 0.05	1.10 ± 0.01	-	3.10 ± 0.03
Lonicerin (9)	0.83 ± 0.01	0.55 ± 0.00	4.53 ± 0.04	0.46 ± 0.00	-	1.81 ± 0.02
Isoquercitrin (8)	1.13 ± 0.01	0.74 ± 0.01	-	0.71 ± 0.01	-	2.50 ± 0.03
3,4-CQA (5)	6.66 ± 0.06	3.40 ± 0.03	5.89 ± 0.05	2.82 ± 0.03	18.46 ± 0.17	9.88 ± 0.09
3,5-CQA (7)	21.38 ± 0.22	26.46 ± 0.27	34.98 ± 0.35	12.25 ± 0.12	11.16 ± 0.11	24.46 ± 0.25
4,5-CQA (10)	7.09 ± 0.02	6.85 ± 0.02	8.19 ± 0.02	6.22 ± 0.02	15.23 ± 0.04	12.19 ± 0.03
3-CQA (2)	21.75 ± 0.33	23.62 ± 0.36	9.42 ± 0.14	26.47 ± 0.40	9.01 ± 0.14	10.63 ± 0.16
5-CQA (1)	1.64 ± 0.02	1.73 ± 0.02	1.63 ± 0.02	5.40 ± 0.07	5.78 ± 0.07	2.79 ± 0.04
4-CQA (3)	2.83 ± 0.02	1.40 ± 0.01	1.42 ± 0.01	3.68 ± 0.03	6.20 ± 0.04	2.71 ± 0.02

* Phenolics are listed with antioxidant capacity declining from the top to the bottom for both DPPH and ABTS assays. ** Numbers within brackets indicate peak orders as listed in Figure 1 and Table 2 and Table 3; - indicates phenolics not detected.

**Table 6 molecules-24-00970-t006:** Correlations among phenolics, antioxidant capacities, and inhibition of DNA damage.

Parameters	TP	TF	DPPH	ABTS	ORAC	IDD	TQP	CA	Lu	Iq	Lo	5-C	3-C	4-C	3,4-dC	3,5-dC	4,5-dC
**Free Phenolics**																	
Total phenols (TP)	1																
Total flavonoids (TF)	0.990 **	1															
DPPH	0.967 **	0.930 **	1														
ABTS	0.996 **	0.996 **	0.955 **	1													
ORAC	0.994 **	0.996 **	0.935 **	0.992 **	1												
Inhibition to DNA damage (IDD)	−0.745	−0.738	−0.825 *	−0.761	−0.711	1											
Total quantified phenolics (TQP)	0.994 **	0.976 **	0.973 **	0.983 **	0.985 **	−0.731	1										
caffeic acid (CA)	0.061	0.040	0.095	0.087	0.022	−0.234	−0.011	1									
Luteoloside (Lu)	0.432	0.508	0.416	0.478	0.454	−0.727	0.407	−0.190	1								
Isoquercitrin (Iq)	0.818 *	0.771	0.846 *	0.799	0.790	−0.725	0.807	0.470	0.160	1							
Lonicerin (Lo)	0.144	0.223	0.154	0.203	0.154	−0.525	0.116	−0.222	0.916 *	−0.172	1						
5-CQA (5-C)	0.870 *	0.906 *	0.723	0.872 *	0.917 **	−0.453	0.851 *	−0.037	0.350	0.660	0.025	1					
3-CQA (3-C)	0.983 **	0.962 **	0.945 **	0.965 **	0.980 **	−0.647	0.992 **	−0.035	0.318	0.802	0.013	0.882 *	1				
4-CQA (4-C)	0.882 *	0.903 *	0.766	0.872 *	0.923 **	−0.525	0.877 *	−0.055	0.361	0.737	−0.005	0.978 **	0.906 *	1			
3,4-diCQA (3,4-dC)	0.779	0.730	0.864 *	0.753	0.747	−0.859 *	0.798	0.174	0.401	0.894 *	0.082	0.542	0.766	0.674	1		
3,5-diCQA (3,5-dC)	0.788	0.732	0.897 *	0.781	0.722	−0.786	0.806	0.025	0.405	0.613	0.301	0.394	0.746	0.426	0.711	1	
4,5-diCQA (4,5-dC)	0.993 **	0.971 **	0.987 **	0.987 **	0.976 **	−0.783	0.990 **	0.122	0.405	0.859 *	0.121	0.818 *	0.973 **	0.843 *	0.825 *	0.828 *	1
**Esterified Phenolics**																	
Total phenols	1																
Total flavonoids	0.997 **	1															
DPPH	0.982 **	0.983 **	1														
ABTS	0.995 **	0.990 **	0.982 **	1													
ORAC	0.995 **	0.992 **	0.969 **	0.997 **	1												
Inhibition to DNA damage	−0.740	−0.749	−0.656	−0.712	−0.752	1											
Total quantified phenolics	0.977 **	0.978 **	0.967 **	0.954 **	0.954 **	−0.762	1										
caffeic acid	0.947 **	0.940 **	0.965 **	0.929 **	0.913 *	−0.630	0.977 **	1									
Luteoloside	0.574	0.584	0.483	0.543	0.589	−0.974 **	0.611	0.470	1								
Isoquercitrin	0.880 *	0.851 *	0.801	0.858 *	0.871 *	−0.642	0.860 *	0.835 *	0.487	1							
Lonicerin	−0.028	0.035	−0.074	−0.040	0.016	−0.273	−0.059	−0.239	0.301	0.835 *	1						
**Insoluble-Bound Phenolics**																	
Total phenols	1																
Total flavonoids	0.997 **	1															
DPPH	0.969 **	0.968 **	1														
ABTS	0.993 **	0.989 **	0.977 **	1													
ORAC	0.993 **	0.996 **	0.945 **	0.979 **	1												
Inhibition to DNA damage	−0.883 *	−0.895 *	−0.846 *	−0.892 *	−0.911 *	1											
Total quantified phenolics	0.870 *	0.895 *	0.779	0.831 *	0.914 *	−0.806	1										
caffeic acid	0.958 **	0.968 **	0.987 **	0.958 **	0.948 **	−0.849 *	0.841 *	1									
Luteoloside	0.795	0.810	0.656	0.730	0.850 *	−0.741	0.947 **	0.709	1								
Isoquercitrin	−0.016	0.029	−0.169	−0.048	0.072	−0.055	0.431	−0.058	0.407	1							

** and * indicate highly significant (*p* < 0.01) and significant (*p* < 0.05) correlations, respectively; note that abbreviations used in the first row are defined in the upper part of the leftmost column.

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
