# Peer review of "Comparison of Free, Esterified, and Insoluble-Bound Phenolics and Their Bioactivities in Three Organs of Lonicera japonica and L. macranthoides"

_molecules, 2019, doi:10.3390/molecules24050970_

Round 1

Reviewer 1 Report

The Manuscript describes a good comparison on free, esterified and insoluble-bound 2 phenolics and their bioactivities in three organs of Lonicera japonica and L. macranthoides, and it is appropriate for publication with minor modifications as described below.

- Line 483. "...P<0.05". Then, it should be inserted: "The Duncan/Dunnet/Tukey test has been used for expression of significant differences with distinct letters next to data values in tables and figures. "

- Figure 2. Indicate the letters of the ANOVA plus Duncan/Dunnet/Tukey test on each bar.

- Table 5. Indicate the letters of the ANOVA plus Duncan/Dunnet/Tukey test next to each data value, as in previous tables.

- Figure 3. Indicate the letters of the ANOVA plus Duncan/Dunnet/Tukey test on each bar.

References

The following recent and relevant references about the topic of the Manuscript, should be included and properly discussed in the Introduction and/or the Results and Discussion sections.

- Han, M.H., Lee, W.S., Nagappan, A., Hong, S.H., Jung, J.H., Park, C., Kim, H.J., Kim, G.Y., Kim, G., Jung, J.M., et al. (2016). Flavonoids Isolated from Flowers of Lonicera japonica Thunb. Inhibit Inflammatory Responses in BV2 Microglial Cells by Suppressing TNF-alpha and IL-beta Through PI3K/Akt/NF-kb Signaling Pathways. Phytother Res 30, 1824-1832.

- Kong, D.X., Li, Y.Q., Bai, M., Deng, Y., Liang, G.X., and Wu, H. (2017). A comparative study of the dynamic accumulation of polyphenol components and the changes in their antioxidant activities in diploid and tetraploid Lonicera japonica. Plant Physiol Biochem 112, 87-96.

- Li, C.Y., Ma, Y., Li, H.Y., and Pen, G.P. (2017). Concentrating Phenolic Acids from Lonicera Japonica by Nanofiltration Technology. In Advances in Materials, Machinery, Electronics I, L. Liu, C. Yang, and J. Ke, eds. (Melville: Amer Inst Physics).

- Mahboob, T., Azlan, A.M., Tan, T.C., Samudi, C., Sekaran, S.D., Nissapatorn, V., and Wiart, C. (2016). Anti-encystment and amoebicidal activity of Lonicera japonica Thunb. and its major constituent Chlorogenic acid in vitro. Asian Pac J Trop Med 9, 844-850.

- Park, K.I., Park, H., Nagappan, A., Hong, G.E., Yumnam, S., Lee, H.J., Kim, E.H., Lee, W.S., Shin, S.C., Kim, J.A., et al. (2017). Polyphenolic compounds from Korean Lonicera japonica Thunb. induces apoptosis via AKT and caspase cascade activation in A549 cells. Oncol Lett 13, 2521-2530.

- Pu, G.B., Zhou, B.Q., and Xiang, F.N. (2017). Isolation and functional characterization of a Lonicera japonica hydroxycinnamoyl transferase involved in chlorogenic acid synthesis. Biologia 72, 608-618.

- Yan, K., Cui, M.X., Zhao, S.J., Chen, X.B., and Tang, X.L. (2016). Salinity Stress Is Beneficial to the Accumulation of Chlorogenic Acids in Honeysuckle (Lonicera japonica Thunb.). Front Plant Sci 7, 10.

- Yan, K., Zhao, S.J., Bian, L.X., and Chen, X.B. (2017). Saline stress enhanced accumulation of leaf phenolics in honeysuckle (Lonicera japonica Thunb.) without induction of oxidative stress. Plant Physiol Biochem 112, 326-334.

Best regards.

Author Response

Reply to Reviewer 1

The Manuscript describes a good comparison on free, esterified and insoluble-bound phenolics and their bioactivities in three organs of Lonicera japonica and L. macranthoides, and it is appropriate for publication with minor modifications as described below.

First, many thanks for your recognition and suggestions on our manuscript. Following are our point-by-point replies to your questions, and some of the questions were replied collectively due to their similarity.

(1)    Point 1: Line 483. "...P<0.05". Then, it should be inserted: "The Duncan/Dunnet/Tukey test has been used for expression of significant differences with distinct letters next to data values in tables and figures."

Reply: We have accepted your suggestion and added these words to Page 18, lines 486-487 ‘The Duncan test has been used for expression of significant differences with distinct letters next to data values in tables and figures.’

(2)    Point 2: Figure 2. Indicate the letters of the ANOVA plus Duncan/Dunnet/Tukey test on each bar.

Point 3: Table 5. Indicate the letters of the ANOVA plus Duncan/Dunnet/Tukey test next to each data value, as in previous tables.

Point 4: Figure 3. Indicate the letters of the ANOVA plus Duncan/Dunnet/Tukey test on each bar.

Reply: We have adopted your suggestions and added the letters of the ANOVA plus Duncan test on each bar in Figure 2 and Figure 3. In Table 5, each data value obtained by multiplying the values from Figure 2 by the proportion of the substance in the total amount, therefore, the ANOVA plus Duncan test can not be carried out.

(3)    Point 5: The following recent and relevant references about the topic of the Manuscript, should be included and properly discussed in the Introduction and/or the Results and Discussion sections.

Reply: Three new references have been added as follows: ‘Therein, phenolics such as luteolin, caffeic acid, caffeoylquinic acid isomers, and dicaffeoylquinic acid isomers have been believed to be the major active ingredients of L. japonica possessing outstanding pharmacological effects such as antibacterial, antioxidant, anti-inflammatory, anti-diabetic, anti-encystment, anti-amoebicidal and anticancer activities [5-11]’. Reference numbers were correspondingly changed thereafter. The last reference section has also been changed.

Reviewer 2 Report

Update introduction and the discussion with the articles:

Gao, Y., Ma, S., Wang, M., & Feng, X. Y. (2017). Characterization of free, conjugated, and bound phenolic acids in seven commonly consumed vegetables. Molecules, 22(11), 1878. doi:10.3390/molecules22111878

Shahidi F., & Ju-Dong Y. (2016). Insoluble-bound phenolics in food. Molecules, 21(9), 1216; doi:10.3390/molecules21091216

Author Response

Reply to Reviewer 2

First, many thanks for your recognition and suggestions on our manuscript. Following are our point-by-point replies to your questions.

(1)    Point 1: 1. Introduction

Lines 65 to 67: insoluble-bound phenolics are covalently bound to cell wall structural component and commonly extracted by alkalization of the residue after the extraction of free and esterified phenolics [13, 14, Shahidi F., & Ju-Dong Y. (2016)]

Page 10 of 22: 2.6. Extraction of Insoluble-Bound Phenolics, in:

Shahidi F., & Ju-Dong Y. (2016). Insoluble-bound phenolics in food. Molecules, 21(9), 1216; doi:10.3390/molecules21091216

Reply: We accept your suggest, and modified as follows: Free phenolics, as well as esterified phenolics that are conjugated to sugars and low-molecular-mass components, are extractable by a solvolytic solution such as water, methanol, ethanol, and acetone, while insoluble-bound phenolics are covalently bound to cell wall structural component and commonly extracted by alkalization of the residue after the extraction of free and esterified phenolics [16-18]. Reference numbers were correspondingly changed thereafter. The last reference section has also been changed.

(2)    2. Results and discussion

Lines 116, 117, 118, 119, 120, 121: Higher proportions of free phenolics fraction in total phenols was also found in leaves of pomelo, longan and litchi [15] and in hyacinth bean, mung bean, pearl bean, Phaseolus calcaratus and Semen Dolichoris Album [24], whereas in black bean, chickpea, cow gram, flower waist bean, kidney bean, red bean, red kidney bean and spring bay bean, contributions of insoluble-bound phenolics were higher than those of free and esterified phenolics fractions [24]. Free, conjugated, and bound phenolic acids of seven commonly consumed vegetables, including kidney bean, cow pea, snow pea, hyacinth bean, green soy bean, soybean sprouts and daylily were characterized [Gao, Y., Ma, S., Wang, M., & Feng, X. Y. (2017)].

Page 11 of 14: seven commonly consumed vegetables from three regions in China (Beijing, Hangzhou, and Guangzhou), were tested for their phenolic acids contents in free, conjugated, and bound forms, in:

Gao, Y., Ma, S., Wang, M., & Feng, X. Y. (2017). Characterization of free, conjugated, and bound phenolic acids in seven commonly consumed vegetables. Molecules, 22(11), 1878. doi:10.3390/molecules22111878

Reply: We accept your suggest, and modified as follows: Higher proportions of free phenolics fraction in total phenols was also found in leaves of pomelo, longan and litchi [19] and in hyacinth bean, mung bean, pearl bean, Phaseolus calcaratus and Semen Dolichoris Album [23], whereas in black bean, chickpea, cow gram, flower waist bean, kidney bean, red bean, red kidney bean and spring bay bean, contributions of insoluble-bound phenolics were higher than those of free and esterified phenolics fractions [23]. Free, conjugated, and bound phenolic acids of seven commonly consumed vegetables, including kidney bean, cow pea, snow pea, hyacinth bean, green soy bean, soybean sprouts and daylily were characterized [24]. Reference numbers were correspondingly changed thereafter. The last reference section has also been changed.

Reviewer 3 Report

Abstract section:

It is suggested a revision of English style.

This section should be rewritten including a more detailed description of antiradical assays.

Table 1, Table 3 and Table 4 Legends: Please correct the typos (P>0.05). Authors know that differences are significant when P<0.05.

Figure 2: Should figure 2 data be identical to those reported in tables, author must delete this figure. Please clarify.

Line 230: The obvious differences between DPPH and ABTS should be substantiated. Particularly, authors should highlight that ABTS is superior to DPPH when applied to samples containing hydrophilic, lipophilic, and highly pigmented antioxidant compounds (Menghini et al. Crocus sativus L. stigmas and byproducts: Qualitative fingerprint, antioxidant potentials and enzyme inhibitory activities. Food Res Int. 2018 Jul;109:91-98. doi: 10.1016/j.foodres.2018.04.028.).

Author Response

Reply to Reviewer 3

First, many thanks for your recognition and suggestions on our manuscript. Following are our point-by-point replies to your questions.

(1)    Point 1: Abstract section:

It is suggested a revision of English style.

This section should be rewritten including a more detailed description of antiradical assays.

Reply: We have checked and improved our manuscript to make the grammar and vocabulary correctly. A more detailed description of antiradical assays has added as follows: ‘Comparisons of three phenolics fractions (i.e., free, esterified and insoluble-bound phenolics) in three different organs (i.e., flower, leaf and stem) of the two species revealed that the free phenolics were the highest in contents of total phenols and total flavonoids, composed of the most numerous phenolics and flavonoids, and thus exhibited the most excellent antioxidant activities (DPPH, ABTS and ORAC) as well as protective effects on DNA damage induced by free radicals’. Because of the limitation of the words, we could not describe them in detail.

(2)    Point 2: Table 1, Table 3 and Table 4 Legends: Please correct the typos (P>0.05). Authors know that differences are significant when P<0.05.

Reply: Thank you for your kind reminder, we have corrected these errors.

(3)    Point 3: Figure 2: Should figure 2 data be identical to those reported in tables, author must delete this figure. Please clarify.

Reply: Figure 2 contains two substances that do not exist in Table 5, i.e., 1,5-diCQA and 1,3-diCQA, for this reason we tend to keep it. The second column in Table 5 contains the same data as Figure 2, so we have deleted them in Table 5.

(4)    Point 4: Line 230: The obvious differences between DPPH and ABTS should be substantiated. Particularly, authors should highlight that ABTS is superior to DPPH when applied to samples containing hydrophilic, lipophilic, and highly pigmented antioxidant compounds (Menghini et al. Crocus sativus L. stigmas and byproducts: Qualitative fingerprint, antioxidant potentials and enzyme inhibitory activities. Food Res Int. 2018 Jul;109:91-98. doi: 10.1016/j.foodres.2018.04.028.).

Reply: We accept your suggest, and modified as follows: And two assays (i.e., DPPH and ABTS assays) were used for assessments of antioxidant capacities of individual phenolics (0.2 mg/ml), including the ten phenolics identified above and two more related dicaffeoylquinic acids (i.e., 1,3-CQA and 1,5-CQA). Particularly, ABTS is superior to DPPH when applied to samples containing hydrophilic, lipophilic, and highly pigmented antioxidant compounds [29].